**An empirical model of the thermospheric mass density derived from CHAMP satellite**

Chao Xiong[1], Hermann Lühr[1], Michael Schmidt[2], Mathis Bloßfeld[2], and Sergei Rudenko[2]

1. GFZ German Research Centre for Geosciences, Telegrafenberg, 14473 Potsdam, Germany.

2. Deutsches Geodätisches Forschungsinstitut at the Technische Universität München (DGFI-TUM), Arcisstr. 21, 80333
Munich, Germany.

Correspondence to: Chao Xiong (bear@gfz-potsdam.de)

**Abstract**

In this study, we present an empirical model, named CH-Therm-2018, of the thermospheric mass density derived from 9-
10   year (from August 2000 to July 2009) accelerometer measurements from the CHAllenging Minisatellite Payload
(CHAMP) satellite at altitudes from 460 to 310 km. The CHAMP dataset is divided into two 5-year periods with 1-year
overlap (from August 2000 to July 2005 and from August 2004 to July 2009) to represent the high-to-moderate and
moderate-to-low solar activity conditions, respectively. The CH-Therm-2018 model describes the thermospheric density
as a function of seven key parameters, namely, the height, solar flux index, season (day of year), magnetic local time,
geographic latitude and longitude, as well as magnetic activity represented by the solar wind merging electric field.
Predictions of the CH-Therm-2018 model agree well with CHAMP observations (within 20%) and show different features
of thermospheric mass density during the two solar activity levels, e.g. the March-September equinox asymmetry and the
longitudinal wave pattern. From the analysis of Satellite Laser Ranging (SLR) observations of the ANDE-Pollux satellite
during August-September 2009, we estimate 6-hour scaling factors of the thermospheric mass density provided by our
model and obtain the median value equal to $1.267 \pm 0.60$. Subsequently, we scale up our CH-Therm-2018 mass density
predicts by a scale factor of 1.267. We further compare the CH-Therm-2018 predictions with the Naval Research
Laboratory Mass Spectrometer Incoherent Scatter Radar Extended (NRLMSISE-00) model. The result shows that our
model better predicts the density evolution during the last solar minimum (2008-2009) than the NRLMSISE-00 model.

**1 Introduction**

The thermosphere is the top layer of the gravitationally bound part of the atmosphere, which is partly ionized and extends
from about 90 km to over 600 km (Lühr et al., 2004). Its density variations are mainly driven by the extreme solar
ultraviolet (EUV) irradiance, the energetic particles and electrical energy from the magnetosphere and solar wind, as well
as by waves originating in the lower atmosphere that propagate upward into the thermosphere. The thermospheric mass
density in general falls off exponentially with increasing altitude, with scale heights of about 25 km to 75 km in the upper
atmosphere, depending on altitude and solar flux levels. In addition to the vertical variation, the mass density varies also
horizontally (latitude and longitude) as well as with solar flux, geomagnetic activity, season and local time (Emmert,
2015).

The thermosphere plays a crucial role for near-Earth space operations, as the total mass density is the key parameter for
orbit perturbation of low Earth orbit (LEO) satellites. Therefore, knowledge of the thermospheric density is critical in the
planning of LEO missions, such as their orbital altitudes, lifetime, and re-entry prediction. As the ionosphere is embedded
in the thermosphere, the knowledge of thermospheric density will also help to improve our understanding of the coupling
between thermosphere, ionosphere and lower-atmosphere (Liu et al., 2013; Emmert, 2015).

There are several tools for measuring the thermospheric mass density. The atmospheric drag provides the most direct
means, which can be measured by onboard accelerometers (e.g., Champion and Marcos, 1973; Lühr et al., 2004;
Doornbos et al., 2010) or estimated from the changes of LEO objects trajectories (e.g., King-Hele, 1987; Emmert at al.,
2004). Other instruments, such as neutral mass spectrometers (e.g., von Zahn, 1970; Hedin, 1983), ultraviolet remote
sensing (e.g., Meier and Picone, 1994; Christensen et al., 2003), as well as the pressure gauge mounted on rockets (e.g.,
The Rocket Panel, 1952; Clemmons et al., 2008), can also be used for inferring the mass density. The details of these

techniques have been reviewed by several earlier studies (e.g., Osborne et al., 2011; Clemmons et al., 2008; Emmert, 2015). Various empirical models have also been developed to describe the thermospheric mass density variability. The most widely used are the Mass Spectrometer Incoherent Scatter Radar Extended (MSISE) model family (Hedin, 1991; Picone et al., 2002), the Drag Temperature Model (Bruinsma et al., 2003, 2012) and the Jacchia-Bowman 2008 (JB2008) model series (Bowman et al., 2008a; Bowman et al., 2008c). Liu et al. (2013) and Yamazaki et al. (2015) reported two empirical models derived from recent LEO missions, such as the CHAllenging Minisatellite Payload (CHAMP, Reigber et al. (2002)) and the Gravity Recovery and Climate Experiment (GRACE, Tapley et al. (2004)). These two models represent well the prominent thermospheric structures at low latitudes like the equatorial mass density anomaly (EMA) and the wave-4 longitudinal pattern, as well as the solar wind influence on the high latitude thermosphere, respectively.

As reported by previous studies, the height and solar activity are the two most important factors that affect the thermosphere mass density (Liu, 2005; Guo et al., 2008; Lei et al., 2012). The CHAMP altitude decreased coincidentally within the declining phase of solar cycle 23. Therefore, it is difficult to fully separate the height and solar activity effects on mass density from CHAMP observations. By assuming a linear dependence on height variation, Liu et al. (2013) used the dataset from 2002 to 2005 when CHAMP was at the altitude of 420 km to 350 km to construct a model, focusing on low- and mid-latitudes. They argued that a linear approximation is applicable within an error of about 3.5% over one scale height. To reduce the height variation effects on the model, Yamazaki et al. (2015) used the MSISE-00 model to normalize the CHAMP and GRACE densities to a common height of 450 km, focusing on high latitudes. However, as the MSISE-00 model was not accurate during the extreme solar minimum of 2008 to 2009 (Thayer et al., 2012; Liu et al., 2014a), it would possibly affect their height correction during the solar minimum period; therefore, they used also the dataset from 2002 to 2006. Both models mentioned above considered only the dataset from high to moderate solar activity, while the dataset from the solar minimum (2008 to 2009) has not been included.

Different to Liu et al. (2013) and Yamazaki et al. (2015), we take into account in this study the dataset from August 2000 to July 2009 for constructing our empirical models of the thermospheric mass density, to make more efficiently use of the CHAMP observation. This period includes high and low solar activities and the CHAMP satellite altitude varies from 450 to 310 km. Both these dependences had not been considered in the aforementioned models. Furthermore, we compare the density results from CHAMP with estimates from a spherical calibration satellite, ANDE-Pollux, which allows us to scale the obtained values to quasi-absolute levels. The rest of the paper is organized as follows. In Sect. 2, we first briefly introduce the CHAMP satellite and its accelerometer measurements, then describe our model construction approach and present the CH-Therm-2018 itself. Our model predictions and the comparison with other models are given in Sect. 3. Section 4 presents a validation of our model using Satellite Laser Ranging (SLR) measurements to the spherical satellite ANDE-Pollux. In Sect. 5 we provide the comparison between our model and the NRLMSISE-00 model. The relevant discussion and summary is given in Section 6.

**2 Data and Model Construction**

**2.1 CHAMP satellite and its accelerometer measurements**

The CHAMP spacecraft was launched on July 15, 2000 into a near-circular polar orbit (inclination: 87.3°) with an initial altitude of 456 km. By the end of the mission, September 19, 2010, the orbit had decayed to about 250 km. For covering all local times, CHAMP needs 131 days. The thermospheric mass density measurements were deduced from the accelerometer onboard CHAMP, which aimed to measure the non-conservative forces exerted on the satellite with a resolution of $<10^{-9}$ m·s$^{-2}$ in along-track and cross-track directions (Reigber et al., 2002). The basic equations for deriving the thermospheric mass density from accelerometer measurements have been described by Lühr et al. (2004) and Liu et al., (2005). And by means of an improved approach the mass density is provided with a resolution of less than $10^{-14}$ kg·m$^{-3}$ (Doornbos et al., 2010). For this study we used the dataset analyzed with the new approach by the Delft group and made available at http://thermosphere.tudelft.nl/acceldrag/data.php.

**2.2 The approach for constructing an empirical model**

To give an overview of the CHAMP mission, Fig. 1 (top panel) shows the satellite altitude variations for the whole mission period. Its mean value decayed from about 460 km in July 2000 to 260 km in September 2010. We see that the satellite was lifted four times (twice in 2002, once in 2006 and 2009) to higher altitude where the air drag is smaller, for extending the lifetime. The thermospheric mass density derived from the on-board accelerometer is presented in the bottom panel, which shows decreasing density from 2002 to 2009, coinciding with the reducing solar flux. But from August 2009 to the end of mission, the derived mass density has increased dramatically from about $5 \cdot 10^{-12}$ to $40 \cdot 10^{-12}$ kg·m$^{-3}$, which is mainly caused by the rapid decrease in satellite altitude during the last mission year but also influenced by the rising activity of the solar cycle 24.

Most important for the variation of thermospheric density is the altitude. In the CH-Therm-2018 model, we consider an exponential dependence on height with a constant scale height for the variation of the mass density. However, as seen in Fig. 1, the CHAMP-measured density has dramatically increased by almost a factor of 8 when its altitude goes below 310 km, which also indicates that a constant scale height is not appropriate for the whole altitude range down to 250 km. Therefore, in this study we consider the 9-year dataset from August 2000 to July 2009 when the satellite was above 310 km, and divide the dataset into two 5-year periods with a 1-year overlap. The two sets of results represent the high-to-moderate and moderate-to-low solar activity conditions, and the altitude of CHAMP decayed from about 460 km to 370 km and from 390 km to 310 km during the two periods, respectively.

The second most important parameter for the mass density variation is the solar flux level. According to Guo et al. (2008), the solar flux index P10.7 is more suitable than F10.7 for characterizing themospheric density variations. P10.7 is defined as P10.7 = (F10.7 + F10.7A)/2, where F10.7A is the 81-day averaged value of the daily F10.7. Fig. 2 (top panel) shows the P10.7 variations from 2000 to the end of 2010, which decreases from over 250 sfu (solar flux unit) in 2002 to below 70 sfu in 2008-2009, and then slightly increases back to 75 sfu at the end 2010. The mean values of P10.7 during the considered two 5-year periods hereafter referred to as P10.7$_{ref}$ are 144.7 and 79.7 sfu, respectively. The bottom panel in Fig. 2 shows the variations of solar wind merging electric field, $E_m$. Liu et al. (2010, 2011) and Zhou et al. (2013) found that $E_m$ is an appropriate parameter to describe the disturbance of the thermospheric mass density by magnetic activities. Considering the memory effect of the magnetosphere-ionosphere-thermosphere system to the solar wind input (Werner and Prölss, 1997; Liu et al., 2010), $E_m$ can be defined as:

$$E_m(t,\tau) = \frac{\int_{t_1}^{t} E_m^{'}(t')e^{(t'-t)/\tau}dt'}{\int_{t_1}^{t} e^{(t'-t)/\tau}dt'} \tag{1}$$

where $E_m^{'}$ represents a continuous function of time $t'$ of the actual merging electric field at the magnetopause. $t_1$ is chosen 3 hours before the actual epoch ($t$), and $\tau$, here 0.5 hours, is the e-folding time of the weighting function in the integrands. For calculating $E_m^{'}$, we use the solar wind to magnetosphere coupling functions, as defined by Newell et al. (2007), and to make $E_m^{'}$ values comparable with the solar wind electric field, the function has been rescaled as:

$$E_m^{'} = \frac{1}{3000} V_{SW}^{\frac{4}{3}}(\sqrt{B_y^{2} + B_z^{2}})^{\frac{2}{3}} \sin^{\frac{8}{3}}(\frac{\theta}{2}) \tag{2}$$

where $V_{SW}$ is the solar wind velocity in km/s and the $B_y$ and $B_z$ in nT are the interplanetary magnetic field (IMF) components in Geocentric Solar Magnetospheric (GSM) coordinates, $\theta$ is the clock angle of the IMF ($\tan(\theta) = |B_y| / B_z$). With these units the value of the merging electric field will result in mV/m. This approach for calculating the merging electric field has also been used by Xiong and Lühr (2014) and Xiong et al. (2016). From Fig. 2 we see that the values of merging electric field are below 5 mV/m during most of the time (slightly higher during higher

solar activity years), with mean values hereafter referred to as $E_{m_{ref}}$ of 1.6 and 1.1 mV/m for the two 5-year periods,

respectively.

Lei et al. (2012) investigated the annual and semi-annual variations of thermospheric density observed by the CHAMP
and GRACE satellites, based on the empirical orthogonal function (EOF) analysis. However, the EOF method does not
consider the physical characteristics, and the basic functions of an EOF-derived model can change significantly by using
different dataset. Therefore, in this study we use the multivariable least-square fitting method for constructing our
empirical model. A similar approach has been applied by Marinov et al. (2004) and Liu et al. (2013). In our model, we
consider the dependences on height (*h*), solar flux (*P10.7*), season (*DoY*, day of year), magnetic local time (*MLT*),
geographic latitude ($\theta$) and longitude ($\phi$), as well as magnetic activity ($E_m$). We use a set of parameters for fitting the
coefficient matrix to the CHAMP measurements, which is expressed as:

$$\rho = f_1(\rho_0, h, H_d) \cdot f_2(P10.7) \cdot f_3(DoY) \cdot f_4(MLT) \cdot f_5(\theta) \cdot f_6(\phi) \cdot f_7(E_m) \qquad (3)$$

where, $\rho_0$ is the mass density at the reference height (310 km, the lowest height of CHAMP during the considered 9-year
period), and $H_d$ denotes the mass density scale height (km). Both parameters are valid for the reference environmental
conditions during the two periods (see below). More discussion of these parameters will follow in Section 4. The seven
sub-functions are defined as:

$$f_1(\rho_0, h, H_d) = \rho_0 \cdot e^{(-(h-310)/H_d)} \qquad (4)$$

$$f_2(P10.7) = a_0 + a_1 \cdot (P10.7 - P10.7_{ref}) + a_2 \cdot (P10.7 - P10.7_{ref})^2 \qquad (5)$$

$$f_3(DoY) = b_0 + \sum_{i=1}^{3} \{b_1(i) \cdot \cos(\frac{i \cdot 2\pi \cdot DoY}{365.25}) + b_2(i) \cdot \sin(\frac{i \cdot 2\pi \cdot DoY}{365.25})\} \qquad (6)$$

$$f_4(MLT) = c_0 + \sum_{j=1}^{4} \{c_1(j) \cdot \cos(\frac{j \cdot 2\pi \cdot MLT}{24}) + c_2(j) \cdot \sin(\frac{j \cdot 2\pi \cdot MLT}{24})\} \qquad (7)$$

$$f_5(\theta) = d_0 + \sum_{k=1}^{6} \{d_1(k) \cdot \cos(\frac{k \cdot 2\pi \cdot \theta}{180}) + d_2(k) \cdot \sin(\frac{k \cdot 2\pi \cdot \theta}{180})\} \qquad (8)$$

$$f_6(\phi) = g_0 + \sum_{l=1}^{4} \{g_1(l) \cdot \cos(\frac{l \cdot 2\pi \cdot \phi}{360}) + g_2(l) \cdot \sin(\frac{l \cdot 2\pi \cdot \phi}{360})\} \qquad (9)$$

$$f_7(E_m) = m_0 + m_1 \cdot (E_m - E_{m_{ref}}) + m_2 \cdot (E_m - E_{m_{ref}})^2 \qquad (10)$$

The height variation of mass density is described by an exponential function, i.e. Eq. (4), and normalized to the altitude at
310 km. To better use the linear and quadratic fitting, P10.7 and $E_m$ have been centered to their mean values (144.7/79.7
sfu and 1.6/1.1 mV/m, respectively) of the two 5-year periods as seen in Eqs. (5) and (10), repsectively. The dependences
of the other parameters, such as season, magnetic local time, geographic latitude and longitude, have been approximated
by trigonometric functions including harmonics from 3 to 6 orders, as shown in Eqs. (6) - (9). In this way 46 parameters
are needed to construct the model, and all the bias values in the Eqs. (5) to (10), namely $a_0$, $b_0$, $c_0$, $d_0$, $g_0$, and $m_0$ have been
set to 1.

**3 CH-Therm-2018 model results**

As described above, by using each 5-year period of CHAMP measurements we have derived empirical models based on
46 free parameters. The values of these parameters are listed in Table 1. Taking all inter-relations into account it results in
a number of 3×3×7×8×12×8×3=145,152 coefficients in our empirical models, both for the high and low solar activity

periods. On top we find the reference density at 310 km altitudes. For the first more active period (mean P10.7=144.7 sfu and $E_m$=1.6 mV/m) we get a value for $\rho_0$ of $7.65 \cdot 10^{-12}$ kg·m$^{-3}$ and for the second low activity period (mean P10.7=79.7 sfu and $E_m$=1.1 mV/m) we get $3.37 \cdot 10^{-12}$ kg·m$^{-3}$. This decrease by a factor of 2.2 reflects primarily the effect of the change in solar flux level. Next in line of Table 1 is the scale height. The derived values of 94 km and 80 km for the two activity periods are quite large. For comparison, Liu et al. (2011) estimated from comparisons of CHAMP and GRACE density measurements scale heights of 83 km and 60 km for solar flux levels of 200 sfu and 80 sfu, respectively. A more detailed discussion of our constant scale height will be given in Section 6.

The obtained dependence of mass density on solar flux level is twice as high during the low solar flux period as during the solar maximum years. This result has to be seen in connection with the obtained scale height. The harmonically varying dependences on season, local time latitude and longitude show no pronounced dependence on the activity level when combining the two amplitudes (cosine and sine) of the fundamental oscillations. Different from that, the relative dependence of the mass density on magnetic activity (parameter at bottom) is significantly higher for low solar activity. In the following we are going to present the main features captured by the two different model solutions.

The panels (a) and (b) of Fig. 3 show the altitude versus solar activity variations from the two periods, over an altitude range from 310 to 470 km. As the level of solar activity is quite different for the two periods, the range of P10.7 has been limited to 100-280 sfu and 65-125 sfu, respectively. The model predicted mass density shows generally similar variations for both periods, which increases with larger solar activity but decreases with altitude. The borders between different colors can be interpreted as constant pressure levels. Panels (c) and (d) of Fig. 3 show the altitude versus geographic latitude variation of the mass density around noon hours. The P10.7 values for the two periods have been set to 150 and 80 sfu, respectively. The mass density generally decreases from low to high latitudes during both periods. For the higher solar activity condition, the equatorial mass density anomaly (EMA), which was earlier described by Liu et al. (2005; 2007) can be seen, with the peak mass density appearing around $\pm 20^o$ latitude. The panels (e) and (f) of Fig. 3 show the dependence of the model predicted mass density on merging electric field during both periods. We see that the mass density increases roughly linearly with the merging electric field and hardly any indication of a saturation effect, which is consistent with results published by Müller et al. (2009) and Liu et al. (2011).

In Fig. 4 the dependence on periodically varying parameters is shown. The panels (a) and (b) present the MLT versus latitude distribution of the mass density. The solar activity has been set again to 150 and 80 sfu for the two periods and the altitude has been set to 400 and 340 km, respectively. During both solar activity periods, the mass density reaches its maximum and minimum around 1400 MLT and 0300 MLT, respectively. The EMA feature is more evident at higher solar activity conditions, as shown in panel (a) of Fig. 4, with larger crest density in the northern hemisphere, as we have chosen predicts for September equinox. Additionally, a clear density trough is seen around -75$^o$ in the southern hemisphere during the lower solar activity conditions. The panels (c) and (d) of Fig. 4 present the seasonal versus latitude variations, showing the mass density peaks at the two equinox seasons and a pronounced minimum around June solstice, which is a well-known feature (e.g. Emmert et al., 2015). An interesting detail is that the mass density exhibits larger amplitudes during the March equinox than during the September equinox for high solar activity condition, while it exhibits an opposite ratio for lower solar activity condition. This equinox asymmetry of thermospheric mass density is consistent with the findings of Liu et al. (2013), who reported that the equinox asymmetry weakens or disappears when the solar flux level falls to below P10.7 = 110 sfu. Guo et al. (2008) argued that the March-September equinox asymmetry can partly be attributed to the inter-annual variability of the thermosphere mass density. Another interesting feature seen from the model predicted result is that at all latitudes the thermospheric mass densities are lower during June solstice than those during December solstice, while the expected hemispheric asymmetry between high-latitude densities during solstice seasons is not evident in our model outputs. We checked the mean annual variations of CHAMP density measurements at various latitude bands, and confirm the dominance of the July minimum at all latitudes with deeper trough in the southern hemisphere.

The coupling between the lower atmosphere and upper atmosphere/ionosphere has been widely investigated in relation to longitudinal wave patterns of different thermospheric/ionospheric parameters (e.g., Immel et al., 2006; Häusler et al., 2007; Liu et al., 2009). The tides excited by latent heat release in tropospheric deep convection tropical clouds can propagate vertically upward (Hagan and Forbes, 2003). These tides vary with season, causing longitudinal patterns with varying wave numbers over the course of a year. Best known are the wave number-4 (WN4) pattern during the months around August and wave number-3 (WN3) pattern around solstice seasons, corresponding to the diurnal eastward propagating DE3 and DE2 tidal components, respectively (e.g., Forbes et al., 2006; Lühr et al., 2008; Wan et al., 2010; Xiong and Lühr., 2013). The panels (e) and (f) of Fig. 4 show the global distribution of the mass density around the noon time for the two considered conditions. Here we find again the EMA signature. Some tidal features, a mixture of longitudinal wave-3 and wave-4 patterns, are found at EMA crest regions in particular during the higher solar activity period. While for the lower solar activity, wave-2 and wave-3 patterns are more prominent. The difference in longitudinal wave patterns may be due to their different wavelengths and their relative susceptibility to molecular dissipation at different solar flux conditions (Bruinsma and Forbes, 2010).

## 4 Density validations by SLR measurements to calibration satellites

So far we have presented density results derived entirely from the CHAMP air drag measurements. Atmospheric drag is the major non-gravitational force acting on LEO satellites, and it causes orbital decay. Since the atmospheric drag depends primarily on the mass density, SLR measurements of spherical LEO satellites can be used to estimate mass density at their altitude. Because of their simple geometry so-called canon-ball satellites can be used for quasi-absolute calibrations. This is not an easy task since, on the one hand, it requires precise modeling of all other gravitational and non-gravitational perturbations acting on the satellites, and on the other hand, the amount of SLR observations contributing globally to LEO satellites observations is low. However, the derived density values can either be used to validate empirical models locally or provide scaling factors for these models (Panzetta et al., 2018).

As an example, we analyzed the SLR observations to the cannon-ball LEO satellite ANDE-Pollux between August 16 and October 3, 2009, and derived from 6-hour to 12-hour time series of estimated scaling factors for the thermospheric density predictions for the CH-Therm-2018 models. Fig. 5 shows the comparison between SLR results and CHAMP estimates in terms of scaling factors. The mean and median values of the derived scaling factors are 1.4 and 1.267, respectively. Also included in the figure is the comparison with the JB2008 model. These values infer that the CH-Therm-2018 model underestimates the thermospheric density at least during the time interval used. In fact, the underestimation of CHAMP density estimates has earlier been suggested by Doornbos (2012), who reported that the CHAMP-derived densities were systematically lower by about 25% than those from GRACE when normalized to a common altitude with the help of an atmospheric model like NRLMSISE-00. Some uncertainty may be introduced by the fact that the ANDE-Pollux observations we compared here are taken from August and September 2009, while the CHAMP dataset we used for the CH-Therm-2018 model ends in July 2009. By taking advantage of the obtained median factor we scaled up all the CH-Therm-2018 predicted mass density values by 1.267.

In addition we compared also the SRL-derived densities with four different empirical models CIRA86 (Hedin et al., 1988), NRLMSISE-00 (Picone et al., 2002), DTM2013 (Bruinsma, 2015) and JB2008 (Bowman et al., 2008c). The corresponding mean values of the estimated scaling factors are 0.65±0.26 for CIRA86, 0.65±0.25 for NRLMSISE-00, 0.79±0.24 for DTM2013 and 0.89±0.27 for JB2008, respectively. It indicates that all these models clearly overestimate the thermospheric density during the period of the low recent solar minimum.

## 5 Comparison with the NRLMSISE-00 model

For reproducing the CHAMP observations with our empirical model, we have combined the results derived from both periods. For the results from August 2000 to July 2004 we use the model predictions from the first 5-year period, while for the results from August 2005 to July 2009 we use the model predictions from the second 5-year periods. For the one-year overlapping period from August 2004 to July 2005, we consider the model predictions from both periods, but use a linearly-weighted combination for the time of overlap. Fig. 6 (top panel) presents our model predictions (red) and

CHAMP observations (black) from August 2000 to July 2009. In general, our model follows quite well the measurements, and even the spikes (corresponding to high magnetic activity) are reasonably well reproduced. For comparison, the middle panel shows also the predictions from the NRLMSISE-00 model (green), which has been divided by the scale factor of 1.267 as derived from Figure 5. Compared to our model, the NRLMSISE-00 model is clearly overestimating the CHAMP observation during solar minimum years. The bottom panel presents quantitatively the relative differences between the model predictions and CHAMP observations:

$$\Delta\rho = \frac{\rho_{model} - \rho_{CHAMP}}{\rho_{CHAMP}} \cdot 100 \tag{11}$$

The annual average differences between our model and observations are within the range ±20% for all nine years, while NRLMSISE-00 overestimates the observations by about 5% for high and moderate solar activity years, and reaches as high as 40% for the extremely low solar activity years. It's no surprise that our model predicts better the observations than the NRLMSISE-00 model, because our model is derived from CHAMP data, which have not been included in the NRLMSISE-00 model.

For a more quantitative inspection of the CHAMP model, we have divided the 9-year dataset into 2-month bins of overlapping 131-day intervals. This time period is required for covering all 24 hours of local time in each bin. For the 2-month bins, we calculate the linear regression slope and the mean ratio between the CHAMP observations and model predictions. The mean ratio is defined as the ratio between the mean values of the observations divided by the model predictions during the 131 days. Examples of this analysis during high (centered on March 1, 2002) and low (centered on November 1, 2008) solar activities are presented in Fig. 7 (a) and (b), respectively. The correlation coefficients between the model predictions and observations reach 0.89 and 0.86, the slopes of the linear fitting are 1.03 and 1.07, and the mean ratios are 1.11 and 1.04. Panel (c) of Fig. 7 presents the slope (top panel) and mean ratio (bottom panel) between the observations and our empirical model (red) as well as the NRLMSISE-00 model (green), respectively. Here again the NRLMSISE-00 model (green) has been downscaled by a factor of 1.267.

The slope CH-Therm-2018 model results vary within the range of 0.6 to 1.2 and the mean ratio varies between 0.9 and 1.2 during almost all the nine years, which are better than those of the NRLMSISE-00 model during the solar minimum (2008-2009). An exception makes the excursion of the slope around 0.6 at the end of 2003. This means both our model and NRLMSISE-00 overestimate the mass density during October and November 2003 (see Fig. 1) a periods of very strong magnetic storms.

It is worth to note that we have extended the model prediction to the last year of the CHAMP mission, as shown in Fig. 7 (c). We see that the slope and the mean ratio between observations and our empirical model have increased dramatically, reaching values of more than 4.0 and 2.0 at the end of the mission, respectively. This is a consequence of the quite low altitude of the CHAMP satellite. Therefore, we have to note that our model is suitable for the altitude range from 310 to 470 km. And the large increase of the CHAMP-measured mass density during the last mission year (see Fig. 1) might be an indication of a smaller scale height due to a composition change at altitudes below 310 km.

## 6 Discussions and Summary

We have constructed a new model of thermospheric neutral density, called CH-Therm-2018, from the CHAMP accelerometer measurements over a 9-year period from August 2000 to July 2009, covering both high and low solar activity conditions (solar flux index P10.7 ranges from over 250 sfu to below 70 sfu). The CHAMP altitude changed from 460 km down to 310 km within this period. Good fits between model and observation are achieved when a constant scale height over this range is assumed. But in addition solar flux level and magnetic activity dependent scaling factors are introduced. This is from the physics point of view not justified because neither the solar flux nor the magnetic activity increases the amount of air particle. Both these parameters change the height distribution of neutral particle and thus modify the scale height. During the CHAMP mission the orbital altitude decreased simultaneously with the reduction of solar flux level. For that reason it is impossible to determine reliably the dependence of the scale height on solar flux from

this dataset. For this modeling purpose this deficiency can be mitigated by a piecewise approximation of the real scale height relation by an exponential function with fixed scale height, and a reference density at 310 km altitude scaled by a solar flux and magnetic activity functions. The two considered periods are 5 years long.

Conventional atmospheric models have often problems with representing the magnetic activity dependence. From Table 1 (bottom rows) it is obvious that the relative dependence on magnetic activity increases significantly when the solar activity goes down. This fact has been noted frequently before. But it is also worth mentioning that the absolute change in mass density with magnetic activity is fairly independent of the solar flux background (see Figs. 3 e and 3f). This confirms earlier claims by Müller et al. (2009) and Liu et al. (2011).

An independent validation of the model-predicted mass densities was performed by comparing with SLR observations on the spherical satellite ANDE-Pollux. Because of the simple geometry of this spacecraft, obtained density estimates can be considered as quasi-absolute. Comparisons performed during the period of low solar activity (August 16 to September 30, 2009), reveals that the density values of the CH-Therm-2018 model should be up-scaled by a factor of 1.267 to fit the SLR observations. This factor has been applied to all model values.

The comparison between our adjusted model predictions with the NRLMSISE-00 model shows that the thermospheric density predicted by the CH-Therm-2018 model agrees well (within $\pm 20\%$) with the CHAMP observations over the whole period, while the NRLMSISE-00 model overestimates the observations by about 40% at the periods low solar activities.

The CH-Therm-2018 model shows quite different features of thermospheric mass density at different solar activity conditions. For example, the EMA feature is more prominent at higher solar activity. The larger density at March equinox than September equinox is only seen at higher solar activity, while this seasonal asymmetry exhibits an opposite sense during lower solar activity conditions. Concerning the tidal signatures at low and equatorial latitudes the thermospheric mass density presents mainly longitudinal wave-4 and wave-3 patterns at higher solar activity, changing to wave-3 and wave-2 patterns at lower solar activity period.

A pending issue for the future studies is a better representation of the mass density height dependence. For this it would be helpful to take simultaneous measurements from at least two satellites into account. Also the extension of the model to lower altitudes, down to the GOCE orbit is planned for a follow-up study.

**Acknowledgements.**

The CHAMP mission was sponsored by the Space Agency of the German Aerospace Center (DLR) through funds of the Federal Ministry of Economics and Technology. The CHAMP thermospheric mass density data are available at the website of air density models derived from multi-satellite drag observations (http://thermosphere.tudelft.nl/acceldrag/data.php). This work is supported by the Priority Program 1788 "Dynamic Earth" of the German Research Foundation (DFG), through the project "Interactions of Low-Orbiting Satellites With the Surrounding Ionosphere and Thermosphere (INSIGHT)".

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

Table 1. The derived values of parameters as defined in Eqs. (4) to (10) for constructing the CH-Therm-2018 empirical model.

| parameters | coefficients | 2000.08-2005.07 | 2004.08-2009.07 |
|---|---|---|---|
| *h* | $\rho_0$ | 7.6540e+00 | 3.3711e+00 |
| | $H_d$ | 9.43487e+01 | 7.99404e+01 |
| *P10.7* | $a_0$ | 1 | 1 |
| | $a_1$ | 9.43396e-03 | 2.08690e-02 |
| | $a_2$ | -2.22615e-06 | -9.76385e-05 |
| *DoY* | $b_0$ | 1 | 1 |
| | $b_{11}$ | 2.09135e-01 | 1.31082e-01 |
| | $b_{12}$ | -1.33610e-01 | -1.18733e-01 |
| | $b_{13}$ | -2.31834e-03 | -4.08388e-02 |
| | $b_{21}$ | 9.57844e-02 | 2.19884e-02 |
| | $b_{22}$ | -4.43634e-02 | -5.93100e-02 |
| | $b_{23}$ | 3.25542e-02 | -1.37226e-02 |
| *MLT* | $c_0$ | 1 | 1 |
| | $c_{11}$ | -2.78983e-01 | -2.77790e-01 |
| | $c_{12}$ | 2.84595e-02 | 3.92145e-02 |
| | $c_{13}$ | -4.49755e-03 | -7.25256e-04 |
| | $c_{14}$ | -9.69936e-03 | 1.52304e-02 |
| | $c_{21}$ | -1.98421e-01 | -2.17354e-01 |
| | $c_{22}$ | 4.30628e-02 | 4.59899e-02 |
| | $c_{23}$ | -9.29224e-03 | 4.73289e-03 |
| | $c_{24}$ | -2.95443e-03 | 1.23554e-02 |
| *θ* | $d_0$ | 1 | 1 |
| | $d_{11}$ | 1.09347e-01 | 1.44814e-01 |
| | $d_{12}$ | -1.29948e-02 | 7.29394e-03 |
| | $d_{13}$ | -8.31644e-03 | -6.45977e-03 |
| | $d_{14}$ | -3.59449e-03 | -1.14291e-03 |
| | $d_{15}$ | 5.22521e-04 | -5.87996e-04 |
| | $d_{16}$ | -1.10054e-03 | 2.19460e-04 |
| | $d_{21}$ | 1.01188e-02 | 5.78031e-02 |
| | $d_{22}$ | 2.34080e-03 | -1.82840e-02 |

|  |  |  |  |
|---|---|---|---|
|  | $d_{23}$ | -9.32401e-04 | 1.23597e-02 |
|  | $d_{24}$ | -1.72102e-03 | -1.22364e-02 |
|  | $d_{25}$ | -1.56578e-03 | 7.92947e-03 |
|  | $d_{26}$ | 1.41373e-03 | -6.42885e-03 |
| $\varphi$ | $g_0$ | 1 | 1 |
|  | $g_{11}$ | -4.77705e-03 | -2.64432e-03 |
|  | $g_{12}$ | -1.47749e-03 | -2.63336e-03 |
|  | $g_{13}$ | 1.51963e-03 | 3.21108e-03 |
|  | $g_{14}$ | 1.65757e-04 | -1.80075e-03 |
|  | $g_{21}$ | -5.66262e-03 | -5.37701e-03 |
|  | $g_{22}$ | 3.01145e-03 | -1.33626e-03 |
|  | $g_{23}$ | 6.08981e-05 | 1.21844e-03 |
|  | $g_{24}$ | 9.34866e-05 | 2.79883e-05 |
| $E_m$ | $m_0$ | 1 | 1 |
|  | $m_1$ | 4.67775e-02 | 1.18627e-01 |
|  | $m_2$ | 3.35777e-04 | -1.36904e-03 |

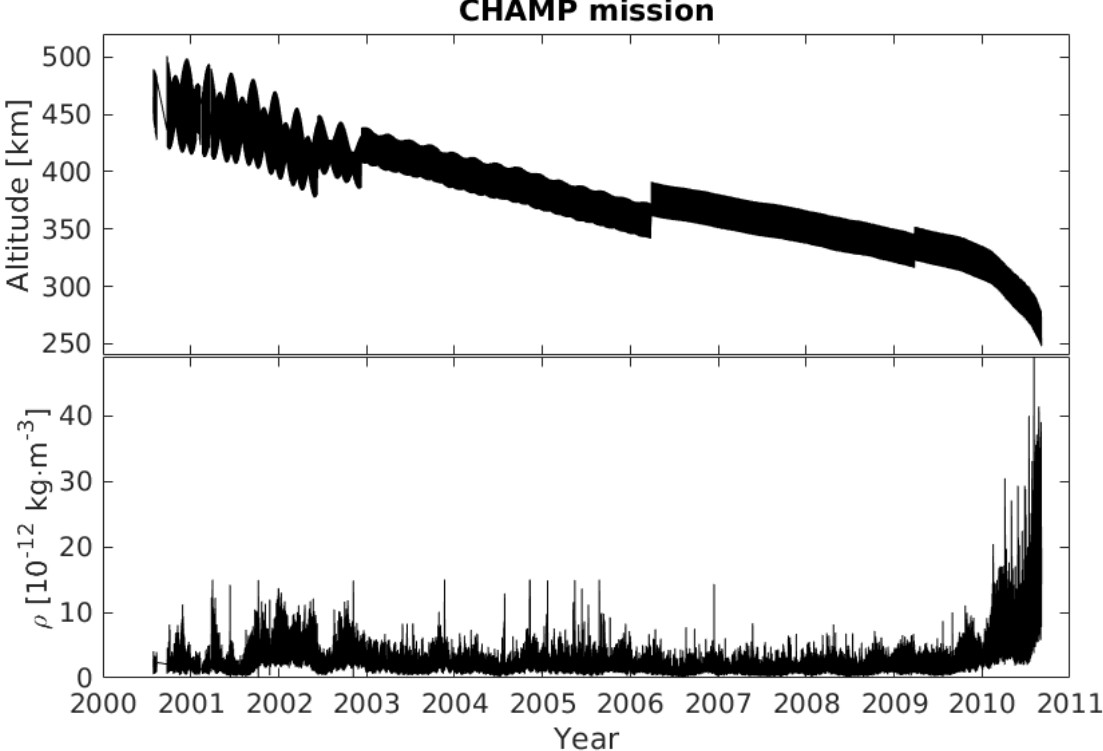

Figure 1. The satellite altitude (top) and thermosphereic mass density (bottom) measured by the CHAMP satellite for the
whole mission period.

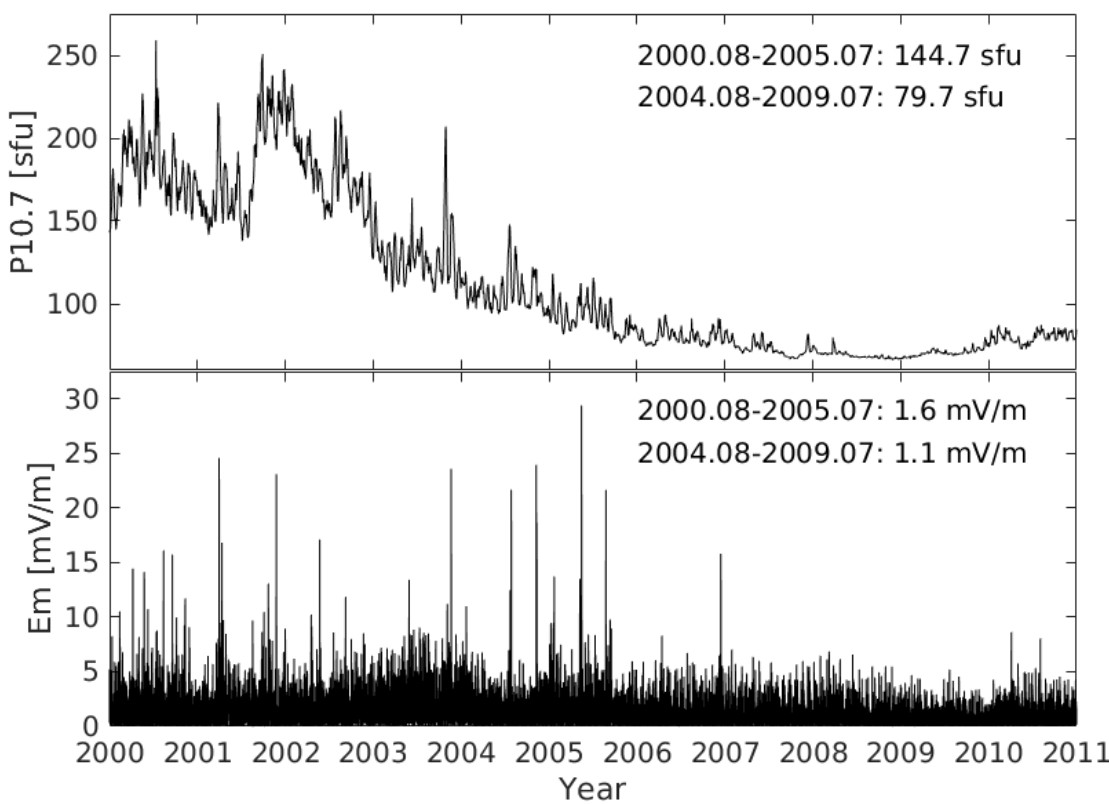

Figure 2. The variations of solar flux index (P10.7, top) and solar wind merging electric field ($E_m$, bottom) from 2000 to
2010. The mean values of two parameters, $P10.7_{ref}$ and $E_{m\ ref}$, during two 5-year periods (from August 2000 to July 2005
and from August 2004 to July 2009, respectively) are given in the upper part of each panel.

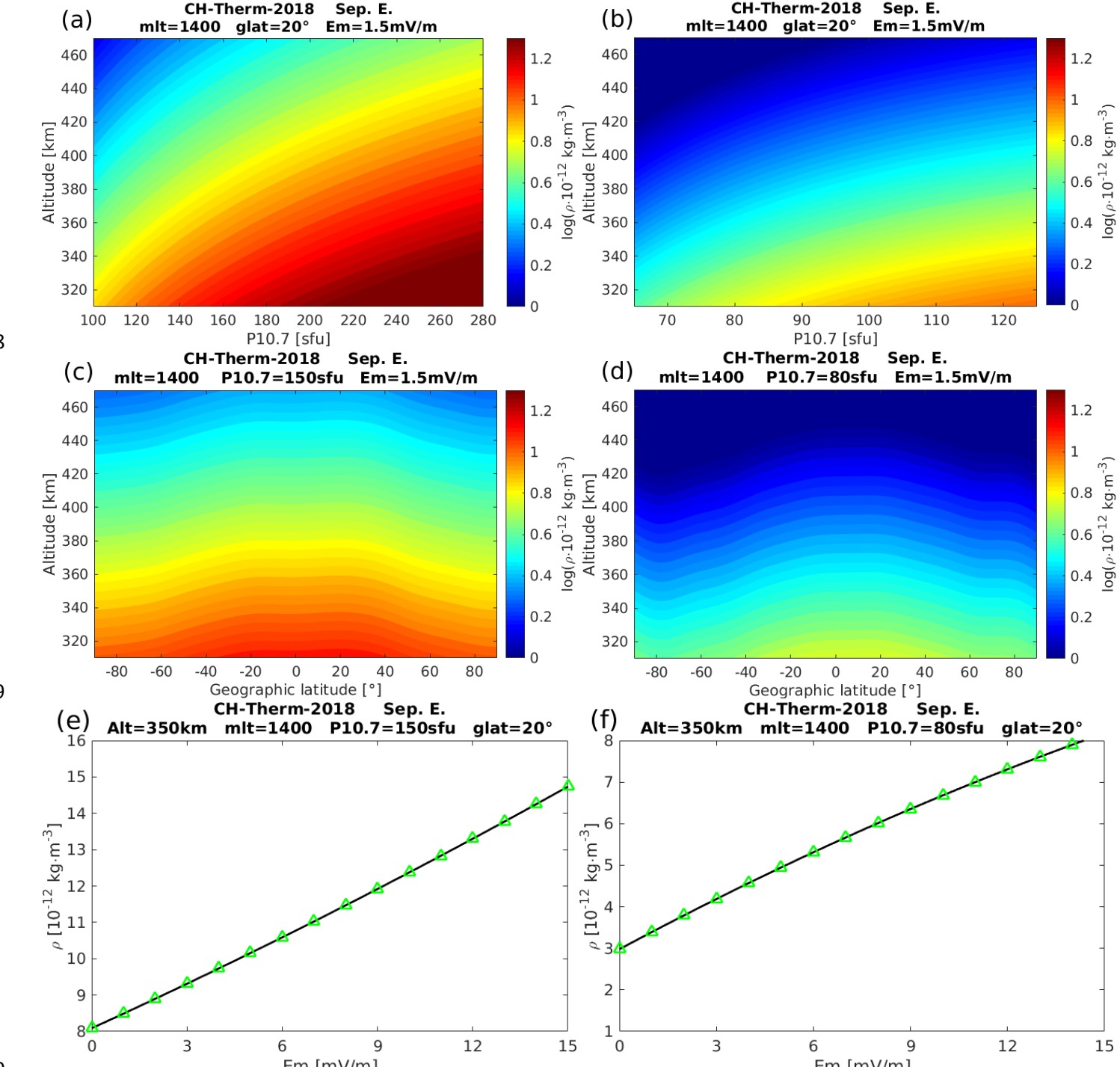

Figure 3. The altitude versus solar activity variations of model-predicted thermospheric mass density around noon at (a)
high and (b) low solar activity conditions. The longitude has been chosen at Greenwich meridian. (c) and (d) are the
altitude versus geographic latitude variations of model predicted mass density for high and low solar activity conditions,
respectively. (e) and (f) shows the dependence of model predicted mass density on merging electric field for both periods.

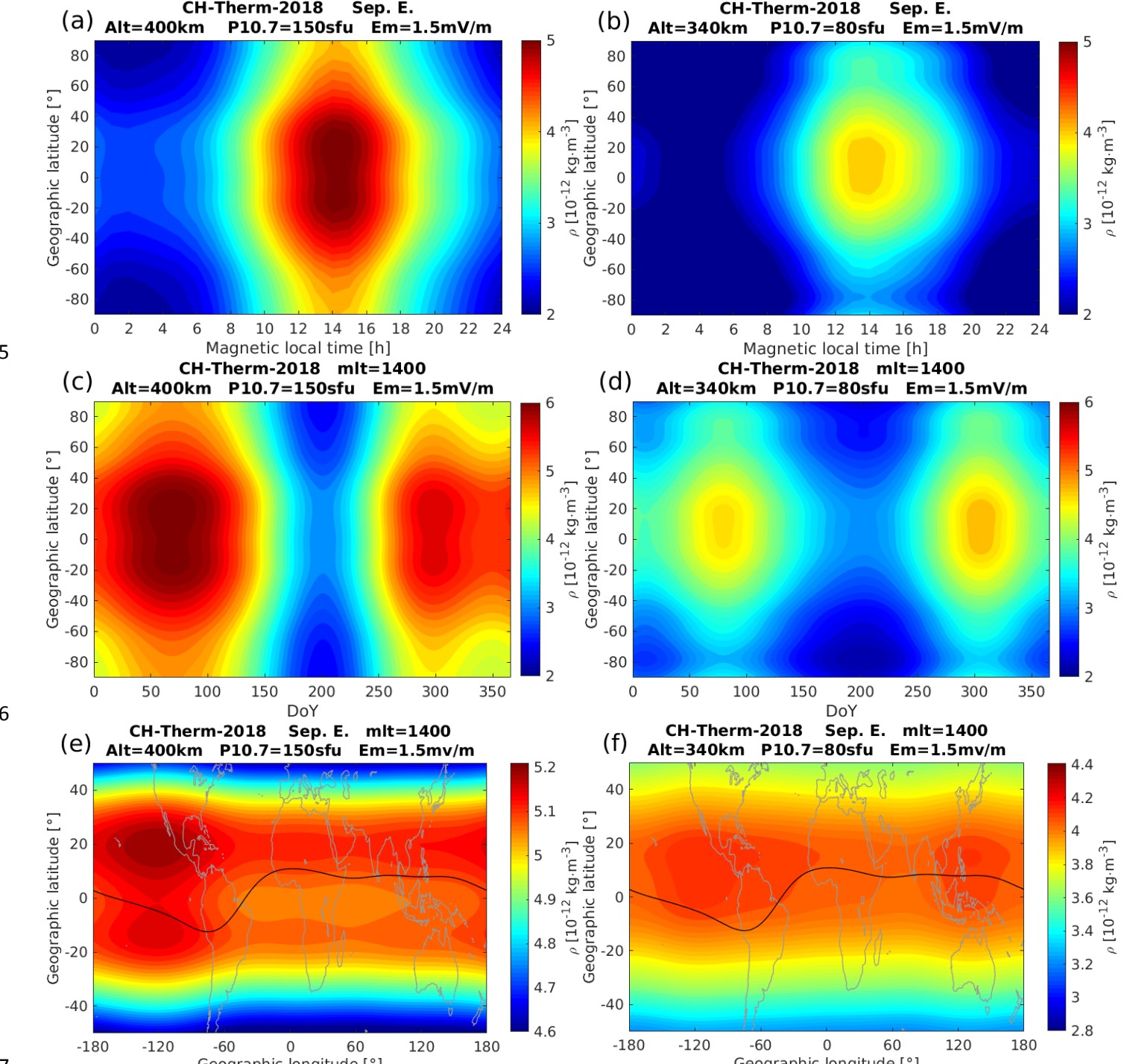

Figure 4. Similar as Figure 3, but for the distribution of (a) and (b): geographic latitude versus magnetic local time; (c) and (d): geographic latitude versus day of year; (e) and (f): geographic latitude versus longitude.

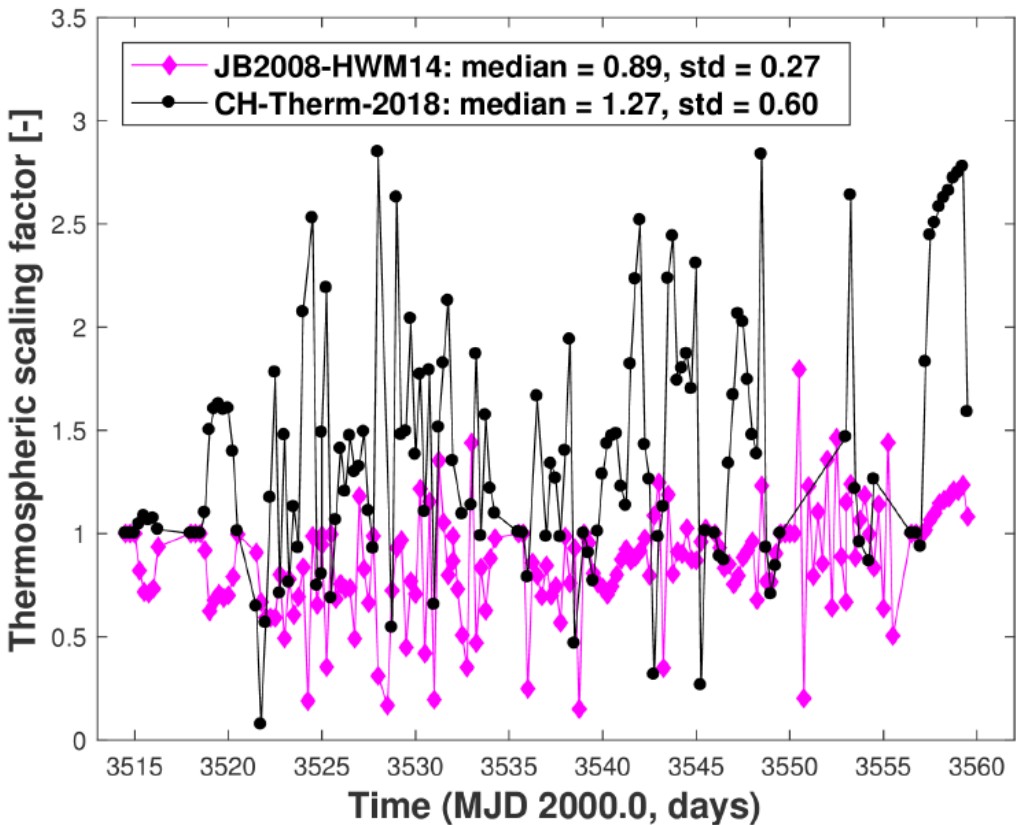

Figure 5. Scaling factors of thermospheric density derived from the analysis of SLR data from the ANDE-Pollux during

August 16 to September 30, 2009 for two models: JB2008 and CH-Therm-2018.

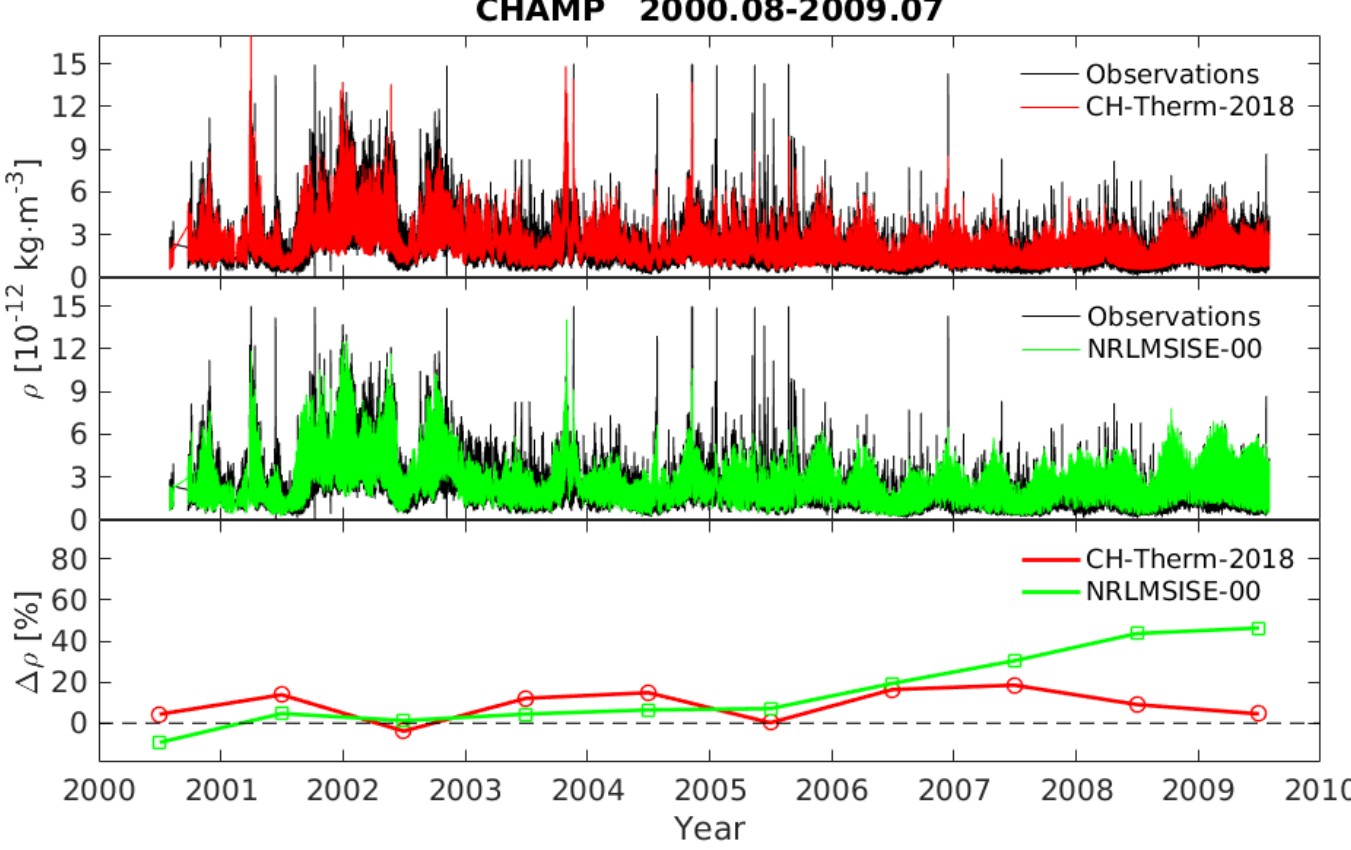

463

Figure 6. The upper panel shows the CH-Therm-2018 model predicted mass density (red) and CHAMP observations
(black) from August 2000 to July 2009. The mid panel shows the same density but for the NRLMSISE-00 model (green)
and CHAMP observations (black). The lower panel gives the annual average relative differences between the model
estimates and CHAMP observations.

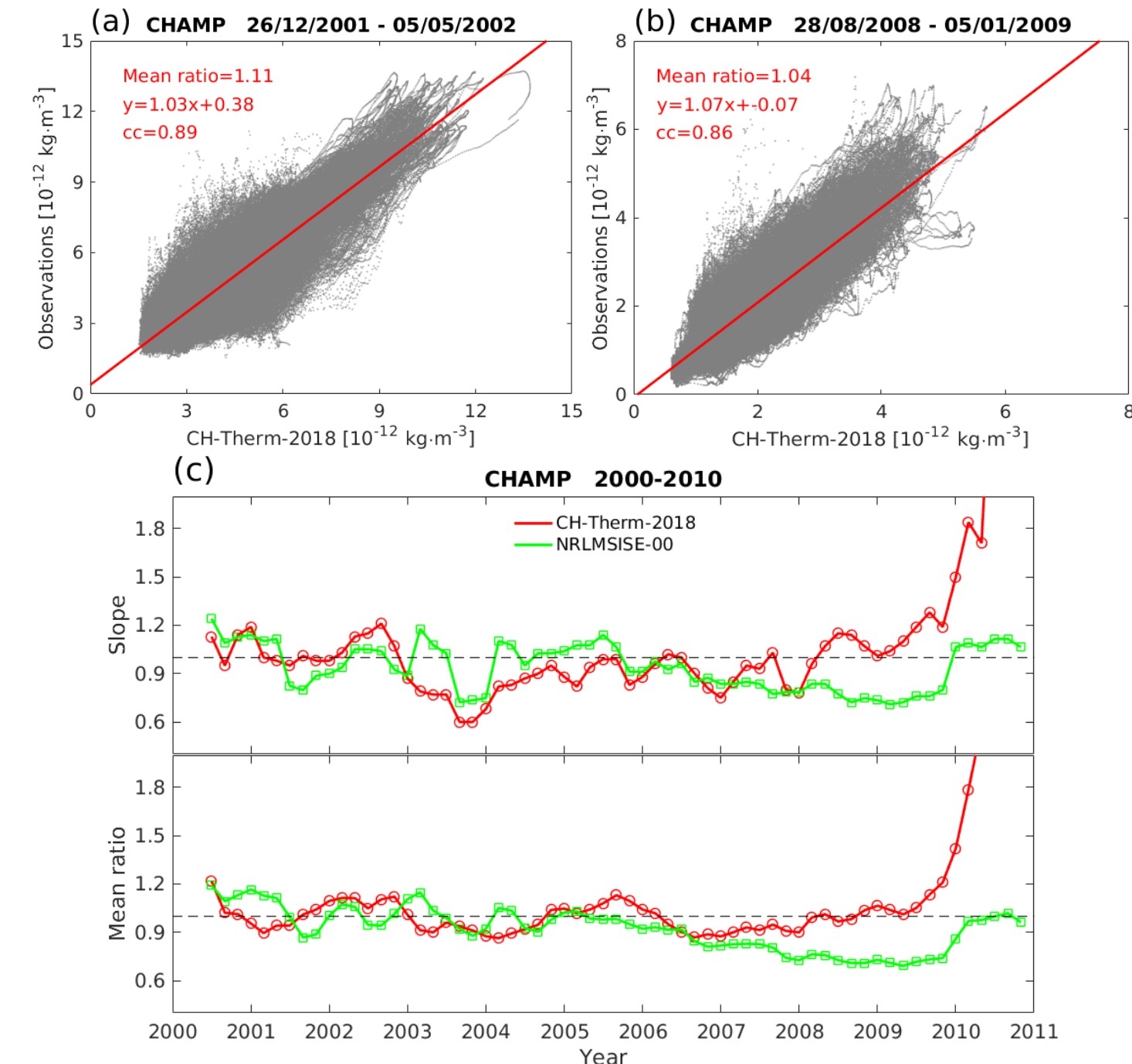

468

469

Figure 7. The linear regression between CHAMP observations and our model predicted results during 131-day period (a) for high (centered on 1 March 2002) and (b) low (centered on 1 November 2008) solar activity conditions, respectively. (c) The red color shows the slope (top panel) and mean ratio (bottom panel) of the linear regression for each 2-month period from 2000 to2010. The green color shows the results from NRLMSISE-00 model.