# Peer review of "An empirical model of the thermospheric mass density derived from CHAMP satellite"

_Annales Geophysicae, 2018_

## Short Comment (SC1) · 26 Apr 2018

Manuscript: angeo-2018-25

Title : An empirical model (CH-Therm-2018) of the thermospheric mass density derived from CHAMP

Authors : Chao Xiong, Hermann Lühr, Michael Schmidt, Mathis Bloßfeld, and Sergei Rudenko

=========================================================================

It is a nice study, which complements previous global mass density studies based on the same CHAMP accelerometer data. This one appears to be built on a simpler set of

functions, e.g., in contrast to empirical orthogonal functions in the papers of Lei et al., 2012 and 2013.

However, I'd like to make some comments and address some critical items of the method used.

First of all, the assumption of a constant scale height in global scale and for all seasons and local times (page 3, bottom paragraph) seems to be unjustified and might lead to apparent abnormal distortions in some of the deduced model parameters. It implies that the neutral temperature is assumed to be constant throughout, while it actually varies at least within a range of factor 2 to 3. The connection (normalisation) to an other empirical model or an iterative approach are a practicable alternative used elsewhere already many times.

It is not explicitly stated in the manuscript - do you use the data set based on the work of Doornbos et al., 2010, or some different approach (page 3, section 2.1)?

The reference height is said to be at 310 km with a fixed (?) mass density "rho_0" of $10^{-12}$ kg/m$^3$ (page 6, line 15). I suppose, it's a guiding or reference mass density. Equations (3) and (4) use the same "rho_0" parameter obviously in a different meaning; the values for the latter are given in Table 1 as ∼0.102 and ∼0.077 for the higher and lower solar activity level, respectively. This should be clarified.

You describe extensively the equinox asymmetry between ∼March and ∼September, but does not mention the annual asymmetry at all, although this is clearly seen as a striking difference between the solstice periods, e.g., in Fig. 4, middle panels, but less obvious as an interhemispheric difference between the December and the June solstice. The missing of the latter might be due to the assumption of the globally constant scale height, mentioned before.

Minors:

page 8, line 9: "depends"

Fig. 7, the insert says "JB2008-HWM14" and gives different numbers of the medium value and the stddev as in the text (page 13, line 5). Is this done here by inclusion of the neutral wind model HWM14? Has the neutral wind been used to correct the mass density (accelerometer) measurements?

Similarly, there is a difference between text and insert with regard to the model CH-Therm-2018 or -2017?

---

## Referee Comment (RC1) · Anonymous Referee #1 · 12 May 2018

In this work, the authors presented an empirical model, named CH-Therm-2018, of the thermospheric mass density derived from 9-year accelerometer measurements at altitude from 460 to 310 km, from CHAMP satellite. This paper is well written, and well organized. However, the referee did not get the point of this study. In other words, I did not see new findings of this work.

Comments:

1. The authors should be addressed the purpose of this work to develop a new empirical model since there are a few models from CHAMP or GRCAE data. Most of the features were mentioned or reported in the previous works, especially in Liu et al. (2013). See more references attached. For me, it is more and less like a student exercise.

2. The CHAMP thermospheric densities derived from different groups show different biases. If the authors did not evaluate these datasets first, the model could be useless.

3. The authors mentioned that they used similar equation as Liu et al. (2013) did. Actually, it is totally different. Liu et al. (2013) used multinomial series, so that they got thousands of coefficients.

4. Although there are 9 year dataset, the data are very sparse if the authors consider so many factors, including latitude, longitude, solar activity, geomagnetic activity, altitude, and so on. How to avoid the overfitting issue? How can use the constant scale height to fit the altitudinal variations without a large dataset?

5. The authors developed two models for low and high solar activities. It is odd for me.

6. It seems that the CH-Therm has a better performance as compared with MSIS. This is expected. When they compared the SLR data, the CH-Therm is even worse than the JB model.

7. "CH-Therm-2018" should be removed from the title.

Ref:

Weimer, D. R., Sutton, E. K., Mlynczak, M. G., & Hunt, L. A. (2016). Intercalibration of neutral density measurements for mapping the thermosphere. Journal of Geophysical Research: Space Physics, 121, 5975–5990. https://doi.org/10.1002/2016JA022691.

Calabia, A., & Jin, S. (2016). New modes and mechanisms of thermospheric mass density variations from GRACE accelerometers. Journal of Geophysical Research: Space Physics, 121, 11, 191–11, 212. https://doi.org/10.1002/2016JA022594.

Ruan et al. (2018). An exospheric temperature model based on CHAMP observations and TIEGCM simulations. Space Weather, 16. https://doi.org/10.1002/2017SW001759

---

## Referee Comment (RC2) · Anonymous Referee #2 · 15 Jun 2018

The work of C. Xiong et al shows a thermospheric empirical model based on the accelerometer measurements of the CHAMP satellite. The analysis looks simple and straightforward. However, in my opinion, a more in-depth reasoning needs to be made because some conditions imposed to the model, might have led to inaccuracy in the presented results.

My main comment is on the evaluation of the height scale factor. Although the altitude of CHAMP shows a strong variability, the authors decided to divide the overall mission in only two periods of 5 year each, one for high and one for low solar activity. Furthermore, the approximation of constant scale height can strongly affect the results, in particular the dependency on the temperature. In my opinion, this part of the methodology should be fully revised.

[Figure]

The work of Liu et al. (2013), cited many times in the paper, shows a more in-depth analysis of the same problem and even better results. As an example, Fig. 6 of Xiong et al. shows a correlation of the CHAMP model wrt to the observations of at most 0.89 in the high solar activity phase, whereas Fig. 1 of the paper of Liu et al., shows a correlation coefficient of 0.96. The choice of a simpler model used in the revised paper is not always understandable.
* * *

---

## Author Comment (AC1) · 9 Jul 2018

Responses to the reviewers' comments on Manuscript angeo-2018-25

**An empirical model (CH-Therm-2018) of the thermospheric mass density derived from CHAMP**

Chao Xiong, Hermann Lühr, Michael Schmidt, Mathis Bloßfeld, and Sergei Rudenko

**#Interactive comment by Dr. Förster, mfo@gfz-potsdam.de**

It is a nice study, which complements previous global mass density studies based on the same CHAMP accelerometer data. This one appears to be built on a simpler set of functions, e.g., in contrast to empirical orthogonal functions in the papers of Lei et al., 2012 and 2013.

Different to Lei et al. (2012; 2013), we use the multivariable least-square fitting method for constructing the empirical model. It is difficult to judge which method is better. Models based on empirical orthogonal function (EOF) analysis do not consider the physical information, and are mainly based on the data. Furthermore, the basic functions of an EOF-derived model can change significantly by changing the data, e.g. by considering a longer or shorter data set. From our point of view as the dependence of the thermospheric mass density on different parameters has been defined before the fitting, the physical motivation for the choice of these functions is much easier to be understood, compared to the empirical orthogonal functions appraoch. Therefore, we use the least-square fitting method in this study. Similar analysis has been performed by earlier studies, e.g., Müller et al. (2009) and Liu et al. (2013).

Müller, S., H. Lühr, and S. Rentz (2009), Solar and magnetospheric forcing of the low latitude thermospheric mass density as observed by CHAMP, Ann. Geophys., 27, 2087–2099, doi:10.5194/angeo-27-2087-2009.

Liu, H., T. Hirano, and S. Watanabe (2013) Empirical model of the thermospheric mass density based on CHAMP satellite observation. J Geophys Res Space Physics, 118, 843–848, doi:10.1002/jgra.50144.

However, I'd like to make some comments and address some critical items of the method used.

Thanks to your valuable comments on our manuscript angeo-2018-25, that will definitely help to improve our results. Below you find our point-by-point reply.

First of all, the assumption of a constant scale height in global scale and for all seasons and local times (page 3, bottom paragraph) seems to be unjustified and might lead to apparent abnormal distortions in some of the deduced model parameters. It implies that the neutral temperature is

assumed to be constant throughout, while it actually varies at least within a range of factor 2 to 3. The connection (normalization) to another empirical model or an iterative approach are a practicable alternative used elsewhere already many times.

We actually started our analysis with a solar flux-dependent scale height, but the resulting fits were disappointing. Satisfying fits between observations and model could only be achieved by introducing a constant scale height (over the height range 310-460 km) as defined in Equation (4) combined with modifications of the reference density by scaling factors. We agree that the scale height actually changes with temperature and composition, which vary, e.g. with solar and magnetic activity, latitude, local time, etc. Therefore, we have selected other six key parameters (defined in Equations 5-10) for describing the variations of neutral density at the reference altitude, 310 km. By using the multivariable least-square fitting method, the variation of scale height depending on different parameters is absorbed by the coefficients in Equations 5-10. As a consequence the coefficient, $H_d$, as defined in Equation (4) can be considered as a mathematical expression for an isothermal atmosphere, but does not reflect the actual scale height. In the revised manuscript we will make these circumstances clearer and discuss the implications.

Different from Yamazaki et al. (2015) and other studies, we did not normalize the measurements to a constant altitude by using estimates from models, as models like MSISE-00 have problems during the extreme solar minimum of 2008-2009 (e.g. Thayer et al, 2012; Liu et al., 2014). It will introduce extra errors into the fitting results. Therefore we decided to allow for height dependence, which worked quite satisfyingly in the end.

Thayer, J., X. Liu, J. Lei, M. Pilinski, and A. Burns (2012), The impact of helium on thermosphere mass density response to geomagnetic activity during the recent solar minimum, J. Geophys. Res., 117, A07315, doi:10.1029/2012JA017832.

Liu, X., J. P. Thayer, A. Burns, W. Wang, and E. Sutton (2014), Altitude variations in the thermosphere mass density response to geomagnetic activity during the recent solar minimum, J. Geophys. Res. Space Physics, 119, 2160–2177, doi:10.1002/2013JA019453.

It is not explicitly stated in the manuscript - do you use the data set based on the work of Doornbos et al., 2010, or some different approach (page 3, section 2.1)?

Yes, we used the same dataset as Doornbos et al. (2010), and we will clarify that in Section 2.1.

The reference height is said to be at 310 km with a fixed mass density "rho_0" of 10ˆ-12 kg/mˆ3 (page 6, line 15). I suppose, it's a guiding or reference mass density. Equations (3) and (4) use the same "rho_0" parameter obviously in a different meaning; the values for the latter are given

in Table 1 as ~0.102 and ~0.077 for the higher and lower solar activity level, respectively. This should be clarified.

The chosen reference height at 310 km is the lowest height of CHAMP considered in the 9-year period. As answered above, our present description is not too clear. The term "rho_0" does not give the density at the reference height, but it is scaled by all the functions as shown in Equations 5-10. By using the multivariable least-square fitting method this one factor for the mass density at 310 km results to 0.102 and $0.077\times10^{12}$ kg·m$^{-3}$ for high and low solar activity level. In the revised discussion we will better explain the physical meaning of the derived parameters.

You describe extensively the equinox asymmetry between ~March and ~September, but does not mention the annual asymmetry at all, although this is clearly seen as a striking difference between the solstice periods, e.g., in Fig. 4, middle panels, but less obvious as an interhemispheric difference between the December and the June solstice. The missing of the latter might be due to the assumption of the globally constant scale height, mentioned before.

We will add the description about the annual variation.

**Minors:**

page 8, line 9: "depends"

Corrected.

Fig. 7, the insert says "JB2008-HWM14" and gives different numbers of the medium value and the stddev as in the text (page 13, line 5). Is this done here by inclusion of the neutral wind model HWM14? Has the neutral wind been used to correct the mass density (accelerometer) measurements?

Similarly, there is a difference between text and insert with regard to the model CH-Therm-2018 or -2017?

Thanks. The insert has been corrected in Figure 7.

---

## Author Comment (AC2) · 9 Jul 2018

**#Referee 1**

In this work, the authors presented an empirical model, named CH-Therm-2018, of the thermospheric mass density derived from 9-year accelerometer measurements at altitude from 460 to 310 km, from CHAMP satellite. This paper is well written, and well organized. However, the referee did not get the point of this study. In other words, I did not see new findings of this work.

Thanks to your valuable comments on our manuscript angeo-2018-25, that will definitely help to improve our results.

**Comments:**

1. The authors should be addressed the purpose of this work to develop a new empirical model since there are a few models from CHAMP or GRCAE data. Most of the features were mentioned or reported in the previous works, especially in Liu et al. (2013). See more references attached. For me, it is more and less like a student exercise.

The method we used in this study is indeed similar to that of Liu et al. (2013), but the difference is that they used only the CHAMP dataset from 2002-2005 (high to moderate solar activity) and focused on low- and middle-latitudes. Yamazaki et al. (2015) used also the CHAMP and GRACE thermospheric mass density to build an empirical model, but they have normalized the observations to a common height of 450 km and only used the data from 2002 to 2006 at high latitudes. Therefore, we want to check if different features result during high and low solar activity levels. Another important fact is that we compare our mass density estimates with results from a cannonball satellite, ANDE-Pollux. These spherically shaped spacecraft are considered as density calibration missions. With the help of that our model is scaled to the "absolute" density level.

These advantages will be spelled out clearer in the revised version.

2. The CHAMP thermospheric densities derived from different groups show different biases. If the authors did not evaluate these datasets first, the model could be useless.

For our analysis we used the data set provided by Doornbos (see Doornbos et al., 2010). This is now explicitly stated in the manuscript. From Figure 5, we see that our model predicts are quite consistent with the observations. For correcting any biases of this data set we made the comparison with the SLR results from the cannon ball satellites, see above.

Doornbos, E., J. Van Den Ijssel, H. Lühr, M. Förster, G. Koppenwallner (2010), Neutral density and crosswind determination from arbitrarily, oriented multiaxis accelerometers on satellites. J Spacecraft Rockets, 47, pp. 580-589.

3. The authors mentioned that they used similar equation as Liu et al. (2013) did. Actually, it is totally different. Liu et al. (2013) used multinomial series, so that they got thousands of coefficients.

We should have been clearer. What we mean is we use a similar parameterization. Still we think the multivariable least-square fitting method is quite similar to that of Liu et al. (2013). If you take the coefficients list from our Table 1 and multiply them, you will get a number of $3\times3\times7\times8\times12\times8\times3= 145152$. This is now clarified.

4. Although there are 9 year dataset, the data are very sparse if the authors consider so many factors, including latitude, longitude, solar activity, geomagnetic activity, altitude, and so on. How to avoid the overfitting issue? How can use the constant scale height to fit the altitudinal variations without a large dataset?

We agree that the 9-year dataset is sparse when compared to the thermospheric variations with altitudes, solar activity, season, etc. Therefore the physical meaning of the different parameters is not obvious. An example for that is our constant scale height (see our answer to Dr. Matthias Förster above). Just the combination of all parameters gives a consistent picture. These facts will be discussed in more details in the revised version.

5. The authors developed two models for low and high solar activities. It is odd for me.

Our parameterization is not capable of tracing the full range of variability over the CHAMP mission. For that reason we decided to divide the dataset into two periods. The most important parameters (altitude and solar activity) vary simultaneously in the CHAMP observations and are therefore challenging to separate. Satisfying agreement with observations is achieved when dividing the data into two periods.

In this way we can check the variability of the mass density with respect to certain parameters for both high and low solar activity periods (see Figs 3 and 4).

6. It seems that the CH-Therm has a better performance as compared with MSIS. This is expected. When they compared the SLR data, the CH-Therm is even worse than the JB model.

The CH-Therm-2018 model performs better than NRLMSIS-00 during the years of the deep solar minimum. In the beginning of the CHAMP mission it covers the same epoch as the underlying data for the NRLMSIS-00. Therefore differences are small.

We regard the systematic difference between our model and the ANDE-Pollux calibration data by 1.267 as a scaling factor for increasing all model values. This value fits quite well the difference between CHAMP and GRACE density estimates, as reported earlier by Doornbos (2011). All the model values are now multiplied by that correction factor before plotting the comparisons. We have restructured the manuscript somewhat, presenting the validation with SLR data earlier in the text, in order to show that it is part of the CH-Therm-2018 model.

It is no surprise that the comparison of JB2008 with the ANDE-Pollux results give a smoother curve. These standard models contain less variability.

Doornbos, E. (2011), Thermospheric density and wind determination from satellite dynamics, Ph.D. Dissertation, 188 pp., University of Delft, available at http://repository.tudelft.nl/.

7. "CH-Therm-2018" should be removed from the title.

Accepted.

Ref: Weimer, D. R., Sutton, E. K., Mlynczak, M. G., & Hunt, L. A. (2016). Intercalibration of neutral density measurements for mapping the thermosphere. Journal of Geophysical Research: Space Physics, 121, 5975–5990. https://doi.org/10.1002/2016JA022691.

Calabia, A., & Jin, S. (2016). New modes and mechanisms of thermospheric mass density variations from GRACE accelerometers. Journal of Geophysical Research: Space Physics, 121, 11, 191–11, 212. https://doi.org/10.1002/2016JA022594.

Ruan et al. (2018). An exospheric temperature model based on CHAMP observations and TIEGCM simulations. Space Weather, 16. https://doi.org/10.1002/2017SW001759

---

## Author Comment (AC3) · 9 Jul 2018

**#Referee 2**

The work of C. Xiong et al. shows a thermospheric empirical model based on the accelerometer measurements of the CHAMP satellite. The analysis looks simple and straightforward. However, in my opinion, a more in-depth reasoning needs to be made because some conditions imposed to the model, might have led to inaccuracy in the presented results.

My main comment is on the evaluation of the height scale factor. Although the altitude of CHAMP shows a strong variability, the authors decided to divide the overall mission in only two periods of 5 year each, one for high and one for low solar activity. Furthermore, the approximation of constant scale height can strongly affect the results, in particular the dependency on the temperature. In my opinion, this part of the methodology should be fully revised.

The work of Liu et al. (2013), cited many times in the paper, shows a more in-depth analysis of the same problem and even better results. As an example, Fig. 6 of Xiong et al. shows a correlation of the CHAMP model wrt to the observations of at most 0.89 in the high solar activity phase, whereas Fig. 1 of the paper of Liu et al., shows a correlation coefficient of 0.96. The choice of a simpler model used in the revised paper is not always understandable.

Thanks to your valuable comments on our manuscript angeo-2018-25, that will definitely help to improve our results.

We indeed used for our analysis a constant scale height in this study as defined in Equation (4), and we agree that the actual scale height depends on many parameters, e.g., altitude, solar activity, latitude, etc. Therefore, we have selected six key parameters (defined in Equations 5-10) for describing the variations of neutral density at the reference altitude, 310 km. By using the multivariable least-square fitting method, the variation of scale height depending on different parameters is absorbed by the coefficients in Equations 5-10. As a consequence the coefficient, $H_d$, as defined in Equation (4) can be considered as a mathematical expression for an isothermal atmosphere, but does not reflect the actual scale height. In the revised manuscript we will make these circumstances clearer and discuss the implications.

The reason why we divided the dataset into two periods is that the CHAMP observations are too sparse when considering all the thermospheric variations with altitude, solar activity, season, etc. Our parameterization is not capable of tracing the full range of variability over the CHAMP mission. For that reason we decided to divide the dataset into two periods. The most important parameters (altitude and solar activity) vary simultaneously in the CHAMP observations and are therefore challenging to separate. Satisfying agreement with observations is achieved when dividing the data into two periods.

In this way we can check the variability of the mass density with respect to certain parameters for both high and low solar activity periods (see Figs 3 and 4).

As for the better performance of the model from Liu et al. (2013), one possible reason is that , they used for Figure 1 only observations from magnetically quiet (ap<32) times at low and middle latitudes (-60°<latitude<60°). But we have considered all the observations at all latitude and get quite good correlation coefficients, 0.89 and 0.86, in our Figure 6. For us the determination of the dependences on magnetic activity is an important aspect. This will be made clearer in the manuscript.